# SCHEMA FOR IN-CONTEXT LEARNING

## ABSTRACT

In-Context Learning (ICL) enables transformer-based language models to adapt to new tasks by conditioning on demonstration examples. However, traditional example-driven in-context learning lacks explicit modules for knowledge retrieval and transfer at the abstraction level. Inspired by cognitive science, specifically schema theory, which holds that humans interpret new information by activating pre-existing mental frameworks (schemas) to structure understanding, we introduce SCHEMA-ACTIVATED IN-CONTEXT LEARNING (SA-ICL). This proposed framework extracts the representation of the Building Blocks of Cognition for the reasoning process instilled from prior examples, creating an abstracted schema — a lightweight, structured template of key inferential steps and their relationships — which is then used to augment a model's reasoning process when presented with a novel question. We demonstrate that a broad range of large language models (LLMs) lack the capacity to form and utilize internal schema-based learning representations implicitly, but instead benefit significantly from explicit schema-based scaffolding. Across chemistry and physics questions from GPQA dataset, our empirical experiment results show that SA-ICL consistently boosts performance (up to 39.67%) when the single demonstration example is of high quality, which simultaneously reduces reliance on the number of demonstrations and enhances interpretability. SCHEMA-ACTIVATED IN-CONTEXT LEARNING not only bridges disparate ICL strategies ranging from pattern priming to Chain-of-Thought (CoT) prompting, but also paves a new path for enhancing human-like reasoning in LLMs.

## 1 INTRODUCTION

In-Context Learning (ICL) has emerged as a dominant approach for adapting large language models (LLMs) to new tasks without requiring fine-tuning or additional parameter updates. By conditioning on a set of demonstrations, ICL enables LLMs to leverage prior knowledge and generalize to unseen examples. Despite its effectiveness, traditional ICL does not align fully with how humans acquire and apply knowledge in real-world learning scenarios, as it lacks mechanisms for episodic memory and context-rich encoding (Li et al., 2024a).

Learning in humans is inherently structured, involving knowledge abstraction, retrieval, and adaptive reasoning. Research in cognitive science suggests that humans develop mental frameworks, called schemas, that organize prior knowledge and facilitate problem solving in new contexts (Rumelhart and Ortony, 1977). These schemas enable efficient retrieval of relevant information and guide interpretation and action, reducing reliance on explicit demonstrations (Rumelhart and Ortony, 1977). Critically, schema activation, consisting of bringing the proper schema into working memory, is essential for effective comprehension and analogical transfer; retrieval alone may not suffice Gick and Holyoak (1983); Gentner (1983).

Recent evidence in the behavior of LLMs mirrors this limitation. For instance, recent models such as GPT-4 retrieve numerous plausible analogs with high recall, but often select incorrect ones due to their reliance on surface-level similarity rather than structural alignment (Puranam et al., 2025). This setback calls for mechanisms that go beyond retrieval, mobilizing schema-like abstractions to guide reasoning.

Inspired by these cognitive insights, we introduce SCHEMA-ACTIVATED IN-CONTEXT LEARNING (SA-ICL), a schema-driven retrieval and reasoning framework for language models. Rather than

retrieving task-specific demonstrations alone, SA-ICL first guides the model to construct a problem schema, which acts as its corresponding structured abstraction. Prior examples are then retrieved using these schemas as keys, and critically, their schemas are then used to activate and refine the schema of the new problem. This activation process enables LLMs to solve problems more efficiently by integrating structured prior knowledge into current reasoning, addressing the structural mapping gap observed in analogical reasoning of LLMs (Puranam et al., 2025). Echoing approaches using latent graph schemas for fast transfer learning (Guntupalli et al., 2023), our method enforces structured abstraction as the medium for retrieval, reasoning, and inference.

A particularly important domain for this approach is scientific reasoning. Although physics and chemistry may appear distinct, their foundational problem-solving strategies often converge on shared relational structures. A prime example is the existence of a "conservation law" schema, a structural template for identifying initial and final states around a core principle. Using schema activation, SA-ICL enables cross-domain transfer, where schemas developed in one scientific field can scaffold reasoning in another, similar to analogical transfer, which depends on mapping hidden relational structures rather than surface similarities (Kang et al., 2025).

We evaluate SA-ICL on the graduate-level scientific benchmark *Graduate-Level Google-Proof Q&A* (GPQA dataset) (Rein et al., 2024), which consists of challenging PhD-level physics, chemistry, and biology multiple-choice questions that require structured reasoning. Our experiments are particularly focused on the physics and chemistry subsets of GPQA dataset. The results show that leveraging SA-ICL enhances accuracy compared to standard ICL (One-Shot) in most scenarios. Most notably, our framework improves accuracy by up to 39.67% over One-Shot for chemistry questions and by up to 34.45% for physics questions, when the retrieved examples are of high similarity. Importantly, we demonstrate that the One-Shot prompting alone does not provide the optimal gain in LLMs' performance during in-context learning, whereas utilizing activated schemas consistently improves reasoning efficiency and effectiveness, especially when the knowledge density is high. We further analyze model outputs to illustrate the interpretability benefits of schema activation.

Our contributions are as follows:

- We propose SCHEMA-ACTIVATED IN-CONTEXT LEARNING (SA-ICL), a novel framework that integrates schema construction, schema-guided retrieval, and schema activation to enable more efficient, generalizable, and interpretable in-context learning.
- We perform comprehensive evaluations comparing SA-ICL with traditional ICL and retrieval-only baselines across multiple scientific reasoning benchmarks, demonstrating consistent improvements in overall accuracy.
- We provide interpretability analyses showing that schema activation facilitates more structured reasoning processes and reduces token reliance, offering a path toward more efficient inference-time reasoning.

Overall, our findings suggest that SA-ICL advances beyond example-driven ICL by bridging retrieval with schema activation, mirroring human cognitive strategies more closely. By leveraging abstract schemas to refine reasoning rather than relying solely on explicit demonstrations, schema-driven ICL reduces dependence on examples and makes inference more efficient and interpretable.

## 2 RELATED WORK

### 2.1 SCHEMA THEORY IN COGNITIVE PSYCHOLOGY

Schema theory is one of the fundamental theories of cognitive psychology. This theory conceptualizes how humans organize and structure knowledge into coherent mental frameworks, or schemas, that are constructed from prior experiences and serve as interpretive structures for understanding new information Rumelhart and Ortony (1977). These abstract structures are dynamic; they actively guide how prior knowledge is encoded and retrieved and how the new information is perceived (Brewer and Treyens, 1981). Classic research in psychology from decades ago, like Bartlett (1932) and Piaget (1952) established that human learning involves either interpreting new information into existing schemas (*assimilation*), or modifying these existing schemas to incorporate novel knowledge (*accommodation*). Activated schemas from the lens of prior knowledge provide a cognitive mechanism that enables efficient problem-solving and reasoning by guiding the retrieval process

and allowing individuals to make inferences and fill in missing details (Anderson and Pichert, 1978; Piaget, 1952). This model of human cognition, where activating the correct abstract structure is key to interpreting a new problem, provides the direct theoretical motivation for the SA-ICL framework.

## 2.2 IN-CONTEXT LEARNING METHODOLOGIES

### 2.2.1 EXAMPLE-DRIVEN APPROACHES (E-ICL)

One-Shot and few-shot learning paradigms (Brown et al., 2020) have been adopted as computationally efficient methodologies (Parnami and Lee, 2022) for enabling language models to perform inference-time in-context learning without requiring internal parameter updates. Example-driven ICL utilizes predefined question-answer pairs to change the probability distribution of output tokens conditioned on user queries and prior knowledge (Wang et al., 2020; Min et al., 2022b).

Despite state-of-the-art (SOTA) LLMs achieving substantially extended context windows, few-shot learning continues to demand extensive computational resources when the number of demonstration samples and their associated token counts increase, resulting in inevitable computational cost inflation (Keles et al., 2023). Furthermore, LLMs demonstrate sensitivity to performance worsening when processing long in-context demonstrations for complex reasoning tasks, making example-driven ICL a compromise for tasks characterized by complex reasoning processes (Li et al., 2024b). While example-driven ICL establishes a connection between human-interpretable prompting and machine learning, LLMs require additional mechanisms to achieve full alignment between their computational processes and human cognitive patterns (Mahowald et al., 2024).

The majority of existing ICL research — including MetaICL and PCW — conceptualizes LLMs primarily as pattern-matching systems operating over prompt examples, without comprehensive analysis of their internal abstraction mechanisms (Min et al., 2022a; Ratner et al., 2023). Current approaches show fundamental limitations, including high task specificity and response rigidity, rather than enabling a generalization across diverse domains or a naturalistic use of real-world knowledge (Yang et al., 2022). Additionally, empirical experiments indicate that traditional example-driven ICL achieves optimal performance only when context lengths extend to hundreds of thousands of tokens through multi-shot prompting (Agarwal et al., 2024). SA-ICL advances the exploration of how language models can generate reasoning processes through their internal knowledge representations in a human-interpretable manner, simultaneously optimizing for performance quality, computational cost, and token efficiency.

### 2.2.2 ABSTRACTION-DRIVEN APPROACHES

While most existing in-context learning methods were heavily example-driven, previous works raised key issues (Saglam et al., 2025; Lampinen et al., 2024; Dong et al., 2022). Recently, the machine learning (ML) community has been witnessing a growing development of in-context learning approaches in a broader perspective (Lampinen et al., 2024), including abstraction-driven in-context learning (A-ICL) (Swaminathan et al., 2023), which could contribute to understanding the way models understand and utilize context. Although previous works have linked the mechanism of induction heads in LLMs to the contextual maintenance and retrieval (CMR) model in human episodic memory (Olsson et al., 2022; Polyn et al., 2009), direct evidence for high-level schema induction in language models remains limited.

In contrast, our work adopts an A-ICL approach that explicitly extracts general reasoning steps, conducting experiments at a higher level than task-specific knowledge. Prior studies have shown that fixed, structured generation — enabled by carefully designed prompting mechanisms or properly constrained decoding strategy — can improve LLMs' performance on reasoning tasks (e.g., ReAct, Program-of-Thoughts) (Yao et al., 2023; Chen et al., 2023). Furthermore, retrieval of previous LLM-generated schemas or demonstrations has been shown to be autoregressively beneficial for ICL, as in prompt-retrieval and retrieval-augmented methods (Rubin et al., 2022; Shi et al., 2024).

### 2.2.3 CHAIN-OF-THOUGHT REASONING

Chain-of-Thought (CoT) reasoning has recently arisen as a critical strategy within ICL by explicitly outlining intermediate steps before arriving at a final answer to enhance the reasoning capabilities (Wei et al., 2022; Kojima et al., 2022), thus significantly improving LLM performance on tasks that

require multi-step inference (Nye et al., 2021; Wang et al., 2023). However, CoT reasoning usually operates within example-driven reasoning frameworks, where explicit reasoning details are provided through a few demonstrations to guide model outputs (Zhang et al., 2023). Various methods have been explored in recent studies to optimize and extend CoT. Multiple thought paths are sampled for reasoning and aggregated before inference to improve output reliability in Self-Consistency Prompting (Wang et al., 2023). Moreover, in Least-to-Most prompting, reasoning is progressively refined by starting with simpler sub-questions (Zhou et al., 2022).

Our proposed framework, SCHEMA-ACTIVATED IN-CONTEXT LEARNING (SA-ICL), although sharing conceptual foundations, constitutes a fundamentally distinct computational paradigm. Traditional CoT prompting operates as an instance-specific methodology wherein the model constructs novel and frequently verbose linear reasoning sequences tailored to each input query, consequently necessitating substantial human engineering of input queries to achieve satisfactory performance on specific tasks (Stechly et al., 2024). The instance-specific trajectories also exhibit limited generalizability across disparate task domains (Stechly et al., 2024). In contrast, SA-ICL incorporates structured schema retrieval mechanisms, enabling language models to access and retrieve abstract, generalized schemas from their prior knowledge, thereby automatically adapting reusable cognitive abstractions to novel problems during inference, rather than depending upon task-specific demonstrations. SA-ICL serves two purposes: enhancing interpretability and facilitating knowledge transfer. These characteristics render SA-ICL particularly effective for complex reasoning tasks requiring high-level conceptual abstraction, including scientific inquiry and hypothesis generation.

## 2.3 OTHER HUMAN-INSPIRED PROMPTING METHODS

Wang and Zhao (2024) applied human introspective reasoning strategies by splitting the question-answer queries into multiple metacognitive prompting steps to improve LLMs' capability in question understanding. This work explores the problem of understanding the gap between human and LLM reasoning processes. Zhou et al. (2023) prompting attempted to address the knowledge loss for LLMs in tasks with chaotic input contexts, where relevant information is obscured by distractors, by guiding LLMs to segment and analyze the input systematically, summarizing the findings as they go, before drawing an answer, to reduce the knowledge loss in long-context scenarios effectively. Retrieval-Augmented Generation (RAG) provides LLMs with access to prior knowledge within a given knowledge base for future queries on similar tasks, which can be considered long-term memories for LLMs. However, traditional RAG limits the quality of the retrieval strategy and the corresponding reasoning logic learned from prior knowledge examples by the quality of existing knowledge base examples. It remains a challenge for existing RAG techniques to adapt to dynamic and interconnected knowledge bases Gutiérrez et al. (2025). **SA-ICL** is built on top of the RAG paradigm and leverages schema theory, which humans use to adapt to the dynamic and interconnected knowledge base, by retrieving abstracted reasoning logic from memorized examples for activating a schema for the new problem. Our work emphasizes using human cognitive schemas to fill in the knowledge gap of LLMs between their perceived examples and similar tasks, where the ground truth answer is not apparent in the input context. In contrast, the knowledge needed is closely related.

## 3 SCHEMA-ACTIVATED IN-CONTEXT LEARNING

We propose this innovative ICL framework, SCHEMA-ACTIVATED IN-CONTEXT LEARNING (SA-ICL), which mimics how humans use previous examples to activate a schema, enabling a better understanding and solution to a new problem. This framework can be applied to any trained large language model and combined with existing prompting techniques. This framework is simple, yet also flexible and extendable, providing a reliable and transparent explanation of how a language model learns from previous examples and turns these abstractions into a powerful schemas that guides it in solving new problems.

### 3.1 OVERALL WORKFLOW

SA-ICL operationalizes schema theory from cognitive science in five steps, aligning abstract schema formation with language model retrieval and reasoning. (i) **Problem Representation**: Given

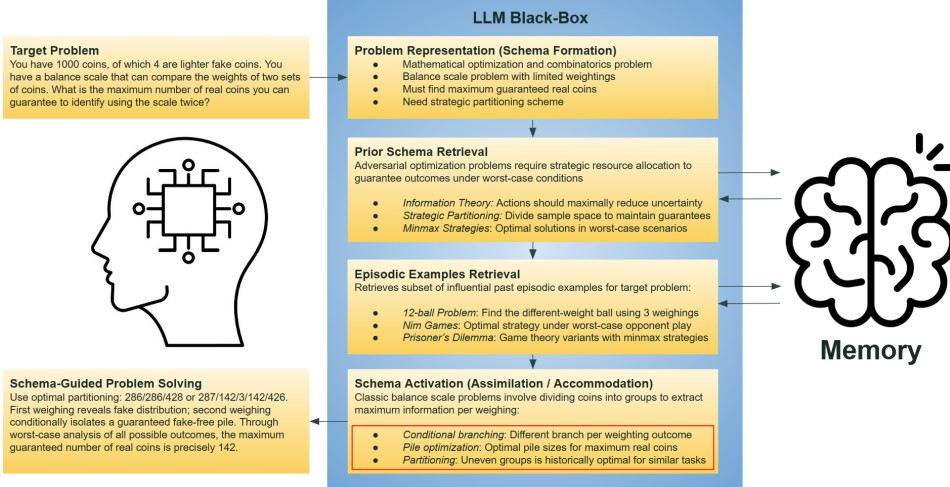

Figure 1: SA-ICL applied to a coin-weighing optimization problem. The framework demonstrates five sequential stages: (i) generates problem representation to form an initial schema recognizing this as a mathematical optimization and partitioning challenge, (ii) retrieves a similar prior schema to identify relevant properties of the question, (iii) gathers a subset of past influential episodic examples that are relevant to solve the target problem, (iv) performs schema activation and integrates retrieved knowledge to develop optimal mathematical strategies adapted to solve the target problem, and (v) conditions the target problem to the adapted schema and utilizes schema-guided inference to eventually conclude that the maximum number of guaranteed real coins is 142.

---

**Algorithm 1** SCHEMA-ACTIVATED IN-CONTEXT LEARNING

---

**Require:** Problem $x$; Schemas $\mathcal{S} = \{\mathcal{S}_1, \ldots, \mathcal{S}_N\}$; Episodic set $\mathcal{E} = \{e_1, \ldots, e_M\}$; Memory $\mathcal{M} = \{(\mathcal{S}_i, e_j, w_{ij}(t))\}$; Representation $\mathcal{R}$; Similarity sim; Activation $f$; LLM; Threshold $\tau \in [0, 1]$

**Ensure:** $y, \mathcal{S}_{\text{new}}$

1: $\mathcal{S}_x \leftarrow \mathcal{R}(x)$
2: $\hat{\imath} \leftarrow \arg\max_{i \in \{1, \ldots, N\}} \ \text{sim}(\mathcal{S}_x, \mathcal{S}_i)$
3: $\hat{\mathcal{S}} \leftarrow \mathcal{S}_{\hat{\imath}}$
4: $\hat{\mathcal{E}}_\tau \leftarrow \emptyset$
5: **for** $j \in \{1, \ldots, M\}$ **do**
6:     **if** $w_{\hat{\imath}j}(t) \geq \tau$ **then**
7:         $\hat{\mathcal{E}}_\tau \leftarrow \hat{\mathcal{E}}_\tau \cup \{e_j\}$
8:     **end if**
9: **end for**
10: $\mathcal{S}_{\text{new}} \leftarrow f(\mathcal{S}_x, \hat{\mathcal{S}}, \hat{\mathcal{E}}_\tau)$
11: $y \leftarrow \text{LLM}(x, \mathcal{S}_{\text{new}})$
12: **return** $(y, \mathcal{S}_{\text{new}})$

---

an input problem $x$, the LLM constructs a representation $\mathcal{S}_x = \mathcal{R}(x)$. This representation functions as the initial schema for the new problem. (ii) **Prior Schema Retrieval**: SA-ICL retrieves the most relevant schema $\hat{\mathcal{S}} \in \mathcal{S}$ that maximizes similarity with $\mathcal{S}_x$. (iii) **Episodic Examples Retrieval**: Conditioned on the retrieved schema $\hat{\mathcal{S}}$, SA-ICL collects a subset of episodic examples whose decayed association weights $w_{\hat{\imath}j}(t)$ exceed a threshold $\tau$. This yields a set $\hat{\mathcal{E}}_\tau$ of examples that remain influential for the current reasoning. (iv) **Schema Activation (Assimilation / Accommodation)**: The retrieved schema $\hat{\mathcal{S}}$ and episodic set $\hat{\mathcal{E}}_\tau$ are integrated with the current problem representation, producing a new activated schema: $\mathcal{S}_{\text{new}} = f(\mathcal{S}_x, \hat{\mathcal{S}}, \hat{\mathcal{E}}_\tau)$. This integration may proceed through *assimilation* when prior schema fits well, or *accommodation* when internal restructuring is required. (v) **Schema-Guided Problem Solving**: Finally, the LLM solves the task by conditioning on the input $x$ and the adapted schema $\mathcal{S}_{\text{new}}$: $y = \text{LLM}(x, \mathcal{S}_{\text{new}})$.

Figure 1 describes the conceptual pipeline of SA-ICL. Section A details the complete mathematical formalization of each step in the framework. Algorithm 1 summarizes SA-ICL framework.

# 4 MAIN EXPERIMENTS

## 4.1 TECHNICAL SET UP

All experiments used standardized OpenAI-style API endpoints. For local runs, we used an NVIDIA A40 GPU with 24GB RAM. In addition to model inference for QWen-3 and Llama-3.1, all embedding generation and reranker computations were also performed on the A40 GPU, with results cached locally for faster loading. Section B summarizes the model families we used and the corresponding execution environments.

## 4.2 TASKS AND METRICS

**Tasks.** We designed closed-ended multiple-choice question-answering tasks wherein language models received individual questions per iteration and applied different reasoning approaches before generating final answers. The experimental design incorporated multiple knowledge density levels to evaluate model performance across varying degrees of prior knowledge acquisition. Closed-ended questions were selected to ensure fair comparison between baseline methods and the SA-ICL.

**Datasets.** The experiments primarily utilized GPQA dataset, a rigorously annotated benchmark containing questions in chemistry and physics that were subjected to comprehensive human annotation. The chemistry subset was employed for initial refinement of schema-based prompting strategies, while the physics subset was also used during the experiment stage. These subsets are designated as *GPQA-Chemistry* and *GPQA-Physics*, respectively. To simulate scenarios with dense knowledge bases. GPT-4o was employed to generate three synthetic variants for each problem in the database through criteria-based prompting (Section C). The criteria defined three distinct similarity levels: *Essentially Same*, *Similar*, and *Different*. The synthetic datasets are designated *GPQA-Chemistry-Synthetic* and *GPQA-Physics-Synthetic*. Section D showed the similarity between the problems in the synthetic datasets and the target problems in the GPQA dataset.

**Evaluation.** For each independent question, the final responses were compared with the ground truths, which led to a downstream performance. We acknowledge that this is an indirect metric.

## 4.3 EXPERIMENTAL SETUP

### 4.3.1 HIGH QUALITY EXAMPLES

The first specific experiment discussed in this paper was a direct response to our research question: *Are examples all we need?* In particular, we investigated the LLMs' performances when the examples were of high quality.

We acknowledge the inherent challenges in curating high-quality exemplars for effective model prompting. To address this, we adopted two distinct strategies:

1. **Synthetic Similarity**: We generated synthetic data using controlled prompts to simulate varying levels of similarity between the generated examples and the target questions (see Section C for detailed prompt templates). We refer to this as *synthetic similarity*.
2. **Latent Similarity**: We used Cohere's Rerank 3.5 to retrieve semantically related examples from GPQA dataset and the synthetic pool, employing cross-encoder rerankers (see Section D for detailed analysis). We refer to this as *latent similarity*.

In this experiment, the LLMs were provided with *Essentially Same* questions as One-Shot examples. We then compare this result with the LLM groups that were provided with schemas. We reported the gaps to answer the question and argue that examples alone were not always sufficient.

### 4.3.2 DENSITY OF KNOWLEDGE BASE ON SCHEMA & DOWNSTREAM PERFORMANCES

To better understand the extent to which SA-ICL depends on the quality of examples when leveraging One-Shot strategies, we conducted systematic experiments by varying the density of examples provided to the LLMs. More specifically, the levels of example relevance from highest to lowest quality for both approaches are listed below:

- **Synthetic Similarity**: *Essentially Same* → *Similar* → *Different*

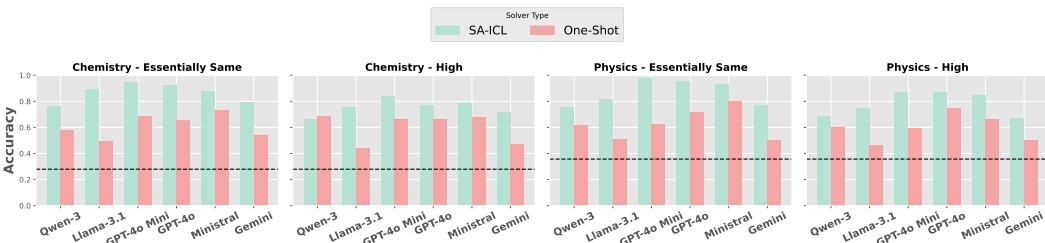

Figure 2: The performances of six LLMs on Chemistry and Physics questions using SA-ICL and example-driven One-Shot prompting, with two retrieval methods (Section 4.3.2). The black dotted line represents the zero-shot performance.

- **Latent Similarity**: *High → Medium → Low*

Notably, we found that a substantial portion of *High* examples correspond to *Essentially Same* (61.18% in chemistry and 67.44% in physics). This experiment investigates both whether SA-ICL is still useful even when the example is of low quality, and when SA-ICL results in the greatest benefits. In our experiments with Latent Simiarity, *High* indicates there is only one example that is of the highest similarity, while *Medium* includes four other examples that had a smaller similarity scores and *Low* includes eight other examples that are less similar to the question compared to the top@1 example.

### 4.3.3 INTERPRETABILITY EXPERIMENTS

As an investigation into the underlying reasoning mechanisms in SA-ICL, we conducted interpretability experiments to analyze decision patterns in model reasoning and confidence in outputs. We selected a representative chemistry problem from *GPQA-Chemistry* and compared three incontext learning techniques: (1) One-Shot + schema, (2) pure One-Shot, and (3) One-Shot + CoT.

For each approach, we extracted token-level probability distributions over multiple-choice options from the model's output logits. This analysis reveals the confidence each method has in arriving at its final answer and looks into whether SA-ICL achieves correct responses through schema activation or surface-level pattern matching. We also analyzed response length and structure to evaluate reasoning efficiency, testing our hypothesis that SA-ICL enables more direct knowledge access compared to the verbose reasoning typically required by CoT approaches.

## 5 MAIN RESULTS

### 5.1 HIGH-QUALITY EXAMPLES DO NOT LEAD TO OPTIMAL PERFORMANCE WITHOUT SCHEMA ACTIVATION

Language models achieved moderate accuracy when only provided with high-quality examples (Section 4.3.2), yet their performance remained suboptimal compared to models employing explicit schema-based learning techniques (Figure 2). This performance gap suggests that relying solely on models' internal representations for high-level abstraction fails to fully exploit the learning potential of quality demonstrations. When models were conditioned to implement schema-based learning through structured guidelines and templates (Algorithm 1), we observed consistently significant improvements in downstream task performance. These performance gains were consistent even when using domain-agnostic schemas without task-specific fine-tuning for physics or chemistry (Sections E and F). We also conducted an experiment with GPT-5 on a different benchmark (Section G).

### 5.2 DENSITY OF KNOWLEDGE AFFECTS THE PERFORMANCE OF SA-ICL

The experimental results presented in Section 5.3 demonstrate a stratified performance profile for the SA-ICL methodology. Under optimal conditions, when the provided exemplar exhibits essential equivalence to the target question, the SA-ICL approach achieves complete dominance with a

perfect win rate against the One-Shot baseline. In sub-optimal conditions characterized by varying degrees of quality and similarity, the SA-ICL method maintains consistent performance advantages, demonstrating superior results across the majority of experimental conditions even when exemplar-question alignment is imperfect.

These findings indicate that while SA-ICL achieves maximum efficacy when provided with highly relevant exemplars, its performance benefits extend beyond ideal matching conditions. The evidence supports the conclusion that SA-ICL constitutes a fundamentally superior prompting methodology rather than a technique dependent solely on exceptional example quality, establishing its viability as a generalizable improvement to existing in-context learning frameworks.

### 5.3 EXTENDING THE EXPERIMENTS TO DATASETS OF MEDXPERTQA, MMLU, AND COMMONSENSEQA

To evaluate the effectiveness of SA-ICL in a broader domains, in particular when the knowledge base is dense. We have conducted evaluations with MedXpertQA (Zuo et al., 2025), MMLU (Wang et al., 2024), and CommonSenseQA (Talmor et al., 2019). We have seen a consistent improvement by SA-ICL as shown in Table 2. We also compared with One-Shot with Reasoning in the context and 5-Shots and observed that SA-ICL could be a more effectiveness context learning approach compared with the CoT Reasoning path and SA-ICL exploits one single best knowledge, reducing the necessity for have multiple-shots, in the dense knowledge scenerios.

### 5.4 ABLATION STUDY: EFFECTIVENESS OF THE ACTIVATION

We conducted an ablation study to investigate whether it is the abstraction of the example or the activation of the schema that boosts LLMs' performance. Since we observed the most significant margin in performance between SA-ICL and One-Shot with the GPT-4o mini, we removed the activation part. We only used the abstraction of the example to solve the Physics questions. We observed that LLMs would not perform as well as when they were allowed to activate the schema Figure I.4. This result highlights the importance of the explicit human-like schema activation.

### 5.5 A DEEPER INVESTIGATION INTO THE EFFECTIVENESS OF SA-ICL

To investigate the interpretable effectiveness of SA-ICL over CoT and One-Shot prompting methods, we logged the log likelihood of the top-5 predicted tokens in the LLM generation output (Figure H.3)(Zhang et al., 2025). One-Shot prompting often encourages the model to overfit to the given demonstration by focusing on fitting the output format (e.g., Answer: ANSWER), rather than leveraging the example to activate prior knowledge and understandings to reason the problem better and provide well-thought-out answers. This problem is discussed in earlier work as pattern-matching.

While prior work noted that enforcing rigid, structured outputs can reduce the flexibility needed for effective reasoning, SA-ICL reached equilibrium between structured reasoning and free-form thinking. Table 3 showed that in high-knowledge settings, where the answer is already well represented in the context, CoT may introduce unnecessary verbosity and even hurt performance, whereas SA-ICL provides more direct and efficient knowledge activation.

## 6 DISCUSSION

SCHEMA-ACTIVATED IN-CONTEXT LEARNING (SA-ICL) challenges the conventional paradigm of machine learning, which has historically relied on large quantities of demonstrations. This counterintuitive shift has constrained the development of abstraction-driven approaches, even in the era of LLMs. However, from a cognitive perspective, SA-ICL demonstrates a stronger alignment with human thinking patterns. Our empirical results demonstrate that activated schemas, viewed through the lens of prior knowledge, enhance the effectiveness of that knowledge when appropriately matched to the problem domain. We demonstrated that schemas enrich the contextual abstraction of examples, and this process facilitates LLMs' understanding of schema generation mechanisms, which subsequently conditions these models to utilize schemas alongside examples more efficiently. In these scenarios, a single example augmented with schema activation yields performance improvements

Table 1: Performances of SA-ICL and One-Shot on Chemistry and Physics questions. For Chemistry, the improvement is up to **39.67%**, **34.88%** for Physics. On average, the improvement in Chemistry is **9.81%**, **12.91%** for Physics. All values in the table were rounded up to the third decimal. Note that for Gemini, the One-Shot, in the Latent Similarity, we were Gemini 2.0 due to the Gemini 1.5 Flash being deprecated.

(a) Chemistry Results

| Model | Method | Synthetic Similarity | | | Latent Similarity | | |
|---|---|---|---|---|---|---|---|
| | | *Essentially Same* | *Similar* | *Different* | High | Medium | Low |
| Qwen-3 | SA-ICL | **0.763** | **0.376** | 0.301 | **0.667** | **0.634** | 0.624 |
| | One-Shot | 0.581 | 0.301 | 0.301 | 0.688 | 0.581 | 0.624 |
| Llama-3.1 | SA-ICL | **0.892** | **0.430** | **0.387** | **0.753** | **0.548** | **0.495** |
| | One-Shot | 0.495 | 0.366 | 0.366 | 0.441 | 0.441 | 0.473 |
| GPT-4o Mini | SA-ICL | **0.946** | **0.462** | **0.366** | **0.839** | 0.581 | 0.559 |
| | One-Shot | 0.688 | 0.366 | 0.323 | 0.667 | **0.624** | **0.613** |
| GPT-4o | SA-ICL | **0.925** | 0.516 | **0.419** | **0.774** | 0.581 | 0.667 |
| | One-Shot | 0.656 | **0.559** | 0.409 | 0.667 | **0.688** | **0.699** |
| Ministral | SA-ICL | **0.882** | **0.473** | **0.376** | **0.785** | 0.634 | 0.624 |
| | One-Shot | 0.731 | 0.376 | 0.280 | 0.677 | **0.656** | **0.656** |
| Gemini 1.5 Flash* | SA-ICL | **0.796** | **0.333** | **0.280** | 0.452 | **0.663** | **0.640** |
| | One-Shot | 0.538 | 0.258 | 0.194 | **0.473** | 0.452 | 0.409 |

(b) Physics Results

| Model | Method | Synthetic Similarity | | | Latent Similarity | | |
|---|---|---|---|---|---|---|---|
| | | *Essentially Same* | *Similar* | *Different* | High | Medium | Low |
| Qwen-3 | SA-ICL | **0.756** | 0.465 | 0.349 | **0.686** | **0.814** | **0.721** |
| | One-Shot | 0.616 | **0.477** | **0.477** | 0.605 | 0.581 | 0.581 |
| Llama-3.1 | SA-ICL | **0.814** | **0.430** | **0.407** | **0.744** | **0.535** | **0.605** |
| | One-Shot | 0.512 | 0.314 | 0.372 | 0.465 | 0.395 | 0.547 |
| GPT-4o Mini | SA-ICL | **0.977** | **0.512** | **0.523** | **0.872** | 0.581 | **0.628** |
| | One-Shot | 0.628 | 0.372 | 0.372 | 0.593 | **0.593** | 0.547 |
| GPT-4o | SA-ICL | **0.953** | **0.663** | **0.616** | **0.872** | 0.674 | 0.698 |
| | One-Shot | 0.721 | 0.616 | 0.547 | 0.744 | **0.698** | **0.721** |
| Ministral | SA-ICL | **0.930** | **0.535** | **0.488** | **0.849** | 0.686 | 0.624 |
| | One-Shot | 0.802 | 0.372 | 0.256 | 0.663 | **0.721** | **0.686** |
| Gemini 1.5 Flash* | SA-ICL | **0.767** | **0.407** | **0.360** | **0.655** | **0.559** | **0.559** |
| | One-Shot | 0.500 | 0.349 | 0.349 | 0.500 | 0.488 | 0.512 |

exceeding 20% compared to using the example alone, indicating that SA-ICL substantially reduces the number of examples required for pattern matching compared to traditional ICL approaches.

It is worth highlighting that although the improvements of SA-ICL are most significant when the episodic examples exhibit high similarity, our framework exhibits noticeable increases in accuracy compared to One-Shot in most scenarios overall. However, there are still special circumstances where accuracy boosts are not observed. We hypothesize this is because our current implementation generates schemas from single examples ($\tau = 1$), which may cause the model to apply reasoning patterns too rigidly without adequate contextual grounding. Nevertheless, through the deployment of the complete SA-ICL algorithm, we posit that dynamic schema activation mechanisms will enhance performance even when the knowledge space exhibits sparsity. We encourage the LLM community to pursue this direction toward developing models with more human-like cognitive capabilities.

Finally, it should be noted that human perception of the environment around them typically does not begin with textual information as the initial sensory input that triggers their inherent schema-based thinking. Instead, visual and other sensory data captured through computer vision systems would likely raise even further the necessity for SA-ICL in real-world deployment scenarios.

Table 2: Performances of SA-ICL, One-Shot, One-Shot with Reasoning and 5-Shots on MedX-pertQA, MMLU, and CommonSenseQA questions. Note that for the MMLU questions were the original MMLU questions included in the MMLU-Pro dataset, for MedXpertQA, we were using the Skeletal questions, for CommonSenseQA, we randomly picked up 200 questions from the validation subset.

| Dataset | Method | Accuracy |
|---------|--------|----------|
| **MMLUCollegeMath** | SA-ICL | **0.7973** |
| | One-Shot | 0.6216 |
| | One-Shot with Reasoning | 0.7162 |
| | 5-Shots | 0.7027 |
| **MedXpertQA** | SA-ICL | **0.7342** |
| | One-Shot | 0.6034 |
| | One-Shot with Reasoning | 0.6667 |
| | 5-Shots | 0.7027 |
| **CommonSense** | SA-ICL | **0.9200** |
| | One-Shot | 0.9000 |
| | One-Shot with Reasoning | 0.8400 |
| | 5-Shots | 0.8800 |

Table 3: Token counts and correctness (✓/✗) across different prompting strategies for the first 10 questions from *GPQA-Chemistry* dataset using GPT-4o Mini with temperature set to 0 for the most consistent results. All prior knowledge in this experiment is retrieved using *High* in Latent Similarity. We ran **three** experiments per question to get the average token counts, and we used the majority correctness as overall correctness.

| Question ID | SA-ICL | | One-Shot | | One-Shot + CoT | |
|-------------|--------|---------|----------|---------|----------------|---------|
| | Tokens | Correct | Tokens | Correct | Tokens | Correct |
| 2662eff7a6231613f...caaeb | 150 | ✓ | 133 | ✗ | 196 | ✗ |
| fc081c2fbb63be500...65420 | 161 | ✓ | 132 | ✓ | 206 | ✓ |
| a8be7a4963bfb6bc7...99122 | 180 | ✓ | 156 | ✓ | 228 | ✓ |
| f730b35adb897658b...a77e5 | 166 | ✓ | 412 | ✗ | 417 | ✗ |
| 1ce3d847d25b2c2f6...01155 | 231 | ✓ | 235 | ✓ | 254 | ✓ |
| d8c36bd55ba561cb4...7a049 | 308 | ✓ | 324 | ✓ | 273 | ✓ |
| 40b2b50a3c993902d...0bfcb | 91 | ✓ | 138 | ✓ | 186 | ✓ |
| a2136b05b78259562...184d7 | 76 | ✓ | 76 | ✗ | 91 | ✗ |
| cbf5c336a0990294b...7d447 | 203 | ✓ | 170 | ✗ | 222 | ✗ |
| 16464cac7090a24d3...9bafd2 | 175 | ✓ | 154 | ✓ | 217 | ✓ |
| **Total Correct** | **10/10** | | **6/10** | | **6/10** | |

## 7 REPRODUCIBILITY STATEMENT

The full code will be posted on GitHub after the review is done. It is worth noting that Gemini 1.5 will be deprecated on Sep 24, 2025. After this date, it will no longer be possible to reproduce the results from our experiments using Gemini 1.5 models. However, the authors will neverthe-less provide full experiment results with Gemini 1.5 to the public. To reproduce the experiments, you can follow the README.md under the zipped code submission in the supplementary mate-rial. The supplementary material also includes the raw experimental results and consists of multiple CSV files for the experiments in Section 4.3, which we used to analyze and report the results in Section 5.3. To examine the raw LLM responses, you can run the following Python command: `base64.urlsafe_b64decode(process_id).decode()`.

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

# A  MATHEMATICAL FORMALIZATION OF SCHEMA-ACTIVATED IN-CONTEXT LEARNING

## A.1  MEMORY: SCHEMAS AND EPISODIC TRACES

We model memory as a bipartite structure linking abstract schemas to multiple episodic examples:

$$\mathcal{M} = \{(\mathcal{S}_i, e_j, w_{ij}(t)) \mid \mathcal{S}_i \in \mathcal{S}, \ e_j \in \mathcal{E}\},$$

where

- $\mathcal{S} = \{\mathcal{S}_1, \ldots, \mathcal{S}_N\}$ is the set of schemas,
- $\mathcal{E} = \{e_1, \ldots, e_M\}$ is the set of episodic traces (examples),
- $w_{ij}(t) \in [0, 1]$ is the association strength between schema $\mathcal{S}_i$ and example $e_j$ at time $t$.

Association weights decay over time, modeling episodic forgetting:

$$w_{ij}(t) = w_{ij}(0) \cdot \exp(-\lambda t), \quad \lambda > 0.$$

The exponential function is an estimate for the forgetting curve, as we want the examples learned earlier to have less impact.

## A.2  PROBLEM REPRESENTATION (SCHEMA FORMATION)

Given an input problem $x$, the LLM constructs a mental representation (schema):

$$\mathcal{S}_x = \mathcal{R}(x)$$

where $\mathcal{R}$ is an embedding or representation function.

## A.3  PRIOR SCHEMA RETRIEVAL

The model retrieves a schema $\hat{\mathcal{S}}$ from $\mathcal{S}$:

$$\hat{\mathcal{S}} = \arg\max_{\mathcal{S}_i \in \mathcal{S}} \text{sim}(\mathcal{S}_x, \mathcal{S}_i),$$

where $\text{sim}(\cdot, \cdot)$ denotes the similarity function, which may be cosine similarity, re-ranking, or another metric.

## A.4  EPISODIC EXAMPLES RETRIEVAL

For notational convenience, let $\hat{i} \in \{1, ..., N\}$ denote the index of $\hat{\mathcal{S}}$ such that $\hat{\mathcal{S}} = \mathcal{S}_{\hat{i}}$. More formally,

$$\hat{i} = \arg\max_{i \in \{1, ..., N\}} \text{sim}(\mathcal{S}_x, \mathcal{S}_i).$$

Afterwards, given the retrieved schema $\hat{\mathcal{S}} = \mathcal{S}_{\hat{i}}$, SA-ICL selects all episodic examples whose (decayed) association to $\hat{\mathcal{S}}$ exceeds a threshold $\tau \in [0, 1]$:

$$\hat{\mathcal{E}}_\tau(t \mid \hat{i}) \ = \ \big\{\, e_j \in \mathcal{E} \ : \ w_{\hat{i}j}(t) \geq \tau \,\big\}.$$

## A.5  SCHEMA ACTIVATION (ASSIMILATION / ACCOMMODATION)

The retrieved schema and its selected episodic set guide activation of a new schema for the current problem:

$$\mathcal{S}_{\text{new}} = f(\mathcal{S}_x, \hat{\mathcal{S}}, \hat{\mathcal{E}}_\tau(t \mid \hat{i})),$$

where $f$ denotes the integration mechanism.

Then, *assimilation* and *accommodation* can be conceptualized as follows:

- **Assimilation:** $\mathcal{S}_{\text{new}} \approx \mathcal{S}_x$ when $\hat{\mathcal{S}}$ fits well.
- **Accommodation:** $\mathcal{S}_{\text{new}}$ requires restructuring when fit is poor.

### A.6 SCHEMA-GUIDED PROBLEM SOLVING

Finally, the LLM produces an output conditioned on the activated schema:

$$y = \text{LLM}(x, \mathcal{S}_{\text{new}}).$$

### A.7 END-TO-END EQUATION

Combining all steps (schema-first, then thresholded episodic selection), we obtain the following equation:

$$y = \text{LLM}\Big(x,\ f\Big(\mathcal{R}(x),\ \underbrace{\arg\max_{\mathcal{S}_i \in \mathcal{S}} \text{sim}(\mathcal{R}(x), \mathcal{S}_i)}_{\hat{\mathcal{S}} = \mathcal{S}_i},\ \underbrace{\big\{\, e_j \in \mathcal{E}\ :\ w_{ij}(t) \geq \tau \,\big\}}_{\hat{\mathcal{E}}_\tau(t|i)}\Big)\Big).$$

## B EXPERIMENTAL SETUP

Table B.1: Experimental setup across model families. "N/A" indicates the parameter count has not been disclosed. Note that embedding and reranker computations were performed on the NVIDIA A40 GPU, with results cached locally.

| Model Family | Parameter Count | Execution Environment |
|---|---|---|
| QWen-3 | 8B | NVIDIA A40 GPU (24GB RAM) |
| LLaMA-3.1 | 8B | NVIDIA A40 GPU (24GB RAM) |
| Ministral | 8B | API endpoint |
| Gemini 1.5 Flash | N/A | API endpoint |
| GPT-4o Mini | N/A | API endpoint |
| GPT-4o | N/A | API endpoint |
| GPT-5 | N/A | API endpoint (subset of experiments) |

## C SYNTHETIC DATASET GENERATION

Below are the prompts that we used to generate the synthetic data that are used as our knowledge base for knowledge and schema retrieval mechanisms. The synthetic data are constructed using GPT-4o via the OpenAI API to ensure the GPQA dataset is not included in the LLM's training data. Specifically, we provide exact prompts for each of the three *synthetic similarity* levels: *Essentially Same*, *Similar*, and *Different*. Note that we intentionally keep all prompts identical to our experiment setup including format, punctuation, and **typos** to ensure reproducibility.

### C.1 *Essentially Same*

```
"""
Consider answer({answer}) and the explanation of solving it({explanation
    }). this question: {question}, along with its

Please generate a new question that is distinct from the previous
    question.

You should follow the following criteria:
- New question requires more knowledge than the provided explanation to
    be used to answer it.
- New question should differ from the given question with a lot of
    distinctiveness.
- Generate a set of new options with only one of them being the correct
    option to the new question.
- Provide three incorrect options, which should be similar to the correct
    answer
```

```
- Provide a short explanation on how to solve the new question, and the
    additional knowledge required to answer the new question.
- Difficulty:
    The new question should be the similar difficulty to the previous
    question.
    If a student has the knowledge to answer the previous question, they
    should have partial knowledge to answer the new question.
    However, the new question should require additional knowledge than
    the given question's scope to be answered.
- Distinctiveness:
    The new question should be distinctive enough to the previous
    question, that the student require additional knowledge to solve the
    problem.
    New question should be unique in its context, and is related to the
    previous question in a minimal level.
- Output Format:
    {question_format}
"""
```

## C.2  *Similar*

```
"""
Consider this question: {question}, along with its answer({answer}) and
    the explanation of solving it({explanation}).

Please give me a slightly different question from this example that test
    the student's ability to transform their knowledge.

You should follow the following criteria:
- The new question only requires the knowledge provided in the
    explanation to be used to answer it.
- New question should still differ with a lot of distinctiveness to test
    student's use of the same knowledge.
- Generate a set of new options with only one of them being the correct
    option to the new question
- Provide three incorrect options, which should be similar to the correct
    answer
- Provide a short explanation on how to solve the new question
- Difficulty:
    The new question should be the similar difficulty to the previous
    question.
    If a student has the knowledge to answer the previous question, they
    should have enough knowledge to answer the new question.
- Distinctiveness:
    The new question should be distinctive enough to the previous
    question, that the student cannot use the same answer.
    New question should be unique in its context, but still related to
    the previous question.
- Output Format:
    {question_format}
"""
```

## C.3  *Different*

```
"""
Consider this question: {question}, along with its answer({answer}) and
    the explanation of solving it({explanation}).

Please generate a new question that is distinct from the previous
    question.

You should follow the following criteria:
```

```
- New question requires more knowledge than the provided explanation to
    be used to answer it.
- New question should differ from the given question with a lot of
    distinctiveness.
- Generate a set of new options with only one of them being the correct
    option to the new question.
- Provide three incorrect options, which should be similar to the correct
    answer
- Provide a short explanation on how to solve the new question, and the
    additional knowledge required to answer the new question.
- Difficulty:
    The new question should be the similar difficulty to the previous
    question.
    If a student has the knowledge to answer the previous question, they
    should have partial knowledge to answer the new question.
    However, the new question should require additional knowledge than
    the given question's scope to be answered.
- Distinctiveness:
    The new question should be distinctive enough to the previous
    question, that the student require additional knowledge to solve the
    problem.
    New question should be unique in its context, and is related to the
    previous question in a minimal level.
- Output Format:
    {question_format}
"""
```

## D  HEATMAPS FOR KNOWLEDGE DENSITY IN LATENT SIMILARITY

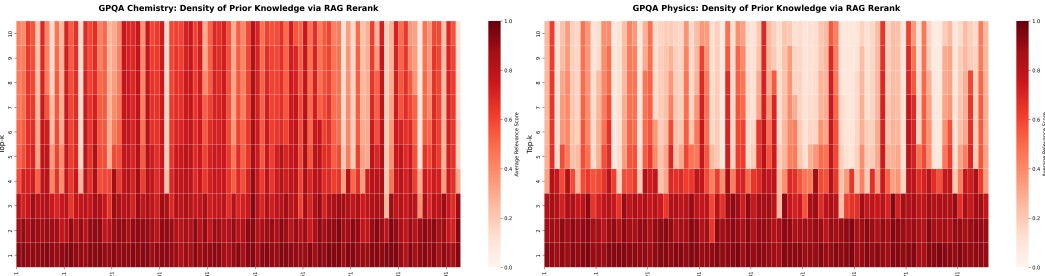

Figure D.1: The heatmaps for knowledge density for GPQA-Chemistry and GPQA-Physics

Knowledge density is defined as the similarity between prior knowledge and a target question. Similarity is quantified by converting prior knowledge into mental representations and employing cross-encoder rerankers to compute relevance scores. The knowledge density heatmaps Figure D.1 visualize marginal relevance scores across top-k retrieved examples for each question, revealing domain-specific patterns in knowledge sparsity. Our analysis demonstrates that chemistry maintains consistently high relevance scores across retrieved examples, while physics exhibits significantly greater sparsity — with relevance scores decreasing 68% compared to chemistry's 40% decrease. These findings underscore that effective schema-based knowledge transfer depends on inherent conceptual coherence within domains rather than universal retrieval mechanisms.

### D.1  LATENT KNOWLEDGE DENSITY METHODOLOGY

We define the latent knowledge density function $\rho : Q \times K \to [0, 1]$, where $Q$ represents the set of questions and $K = \{1, 2, ..., k_{max}\}$ denotes retrieval ranks. In our case, $k_{max} := 10$. For each question $q \in Q$ and rank $k \in K$, $\rho(q, k)$ measures the relevance score between $q$ and its $k$-th most similar prior example, computed using Cohere's Rerank 3.5 cross-encoder architecture. The aggregate density metric $\bar{\rho}(q) = \frac{1}{k_{max}} \sum_{k=1}^{k_{max}} \rho(q, k)$ quantifies overall knowledge availability for

question $q$, where higher values indicate robust transfer potential and lower values reveal knowledge sparsity. We define $\sigma \in [0, 1]$ as the relevance threshold hyperparameter to quantify what we consider quality knowledge. For this investigation, we arbitrarily set $\sigma := 0.5$.

## D.2 HEATMAP VISUALIZATION AND INTERPRETATION

Figure D.1 visualizes $\rho(q, k)$ for GPQA-Chemistry and GPQA-Physics datasets as heatmaps, where each cell $(i, j)$ represents $\rho(q_i, k_j)$ with $q_i$ denoting the $i$-th question (horizontal axis) and $k_j \in \{1, 2, ..., k_{max}\}$ the retrieval rank (vertical axis). Color intensity maps linearly to $\rho$ values, from light shades ($\rho \to 0$) to dark red ($\rho \to 1$), symbolizing low to high relevance scores respectively.

Chemistry questions maintain $\rho_{chem}(q, k) > \sigma$ for most $(q, k) \in Q \times K$ pairs even at $k = k_{max}$, indicating persistent semantic similarity with small $\Delta\rho = \rho(q, k) - \rho(q, k + 1)$ for all $k \in \{1, 2, ..., k_{max} - 1\}$. Physics exhibits rapid density decay with $\rho_{phys}(q, k) < 0.3$ for many questions by $k = 5$, revealing sparse knowledge availability beyond initial retrievals.

## D.3 DOMAIN SPARSITY AND KNOWLEDGE SPACE ISOLATION

Defining decay rate as $\theta(k) = \frac{\rho(q,k)}{\rho(q,1)}$, we observe $\bar{\theta}_{chem}(k_{max}) = 0.60$ for chemistry versus $\bar{\theta}_{phys}(k_{max}) = 0.32$ for physics, where the bar notation indicates domain averaging. This represents $\bar{\rho}_{chem}(q, k_{max}) \approx 0.6 \cdot \bar{\rho}_{chem}(q, 1)$ netting a 40% decrease, compared to $\bar{\rho}_{phys}(q, k_{max}) \approx 0.32 \cdot \bar{\rho}_{phys}(q, 1)$ which yields a 68% decrease. Chemistry maintains $\bar{\rho}_{chem}(q, k) > \sigma$ throughout all $k \in \{1, 2, ..., k_{max}\}$, while physics shows a domain-specific density gradient $\nabla_k \rho$ approximately twice as steep.

We hypothesize that this disparity stems from the inherent conceptual fragmentation of physics across fields like quantum mechanics and general relativity — each possessing specialized theoretical vocabularies that create isolated knowledge regions within the latent space. The intercluster density satisfies $\rho(q_i, e_j) \ll \sigma$ when $q_i$ and the episodic trace $e_j$ belong to different physics subdomains. For example, specialized concepts inherent to quantum mechanics, such as quantum entanglement and wave functions, yield $\rho \approx 0$ when matched against general relativity examples, forming disjoint clusters and thus leading to poorer schema activations.

These findings demonstrate that effective schema-based knowledge transfer depends critically on domain density structure $\rho(q, k)$. High-density regimes where $\bar{\rho} > \sigma$ support extensive retrieval sets, while sparse domains with rapidly decaying $\rho(q, k)$ require adaptive mechanisms for $\rho \to 0$ when $k > k_{threshold}$.

# E DOMAIN-AGNOSTIC SCHEMA TEMPLATE

When generating mental representations or schemas, LLMs were guided for each attribute of the schema.

```
"""Drawing on schema theory from cognitive psychology, think about a high
    -level abstraction (schema) of the problem to guide your reasoning.
    Your ultimate goal is to select the most appropriate answer.:

Below is the template for the schema you need to fill out:

Broad Category:
Identify the overarching subject and general category to which the
    problem belongs.

Refinement:
Describe further details or specific aspects that narrow down the broad
    category.

Specific Scope:
Define the precise focus or context of the problem within the refined
    category.
```

```
Goal:
Clearly state the objective or intended outcome of solving the problem.

Finally, summarize the schema in a few sentences to help students grasp
    the key points. The problem you need to abstract is as follows:"""
```

The LLM will then generate a JSON-like object that represents a concrete schema for a specific question:

```
{
    "schema": {
        "broad_category": str,
        "refinement": str,
        "specific_scope": str,
        "goal": str
    },
    "summary": str
}
```

## F    A FULL CONVERSATION HISTORY DURING THE INFERENCE TIME WITH GPT-4O MINI

Note that all three methods use the same dynamic response JSON format:

```
{
  "name": "DynamicResponse",
  "strict": true,
  "schema": {
    "$defs": {
      "AnswerEnum": {
        "enum": [
          "14",
          "12",
          "10",
          "11"
        ],
        "title": "AnswerEnum",
        "type": "string"
      }
    },
    "properties": {
      "reasoning": {
        "title": "Reasoning",
        "type": "string"
      },
      "final_answer": {
        "$ref": "#/$defs/AnswerEnum"
      }
    },
    "required": [
      "reasoning",
      "final_answer"
    ],
    "title": "DynamicResponse",
    "type": "object",
    "additionalProperties": false
  }
}
```

### F.1    SCHEMA-ACTIVATED IN-CONTEXT LEARNING

System Message:

```
"""Drawing on schema theory from cognitive psychology, think about a high
    -level abstraction (schema) of the problem to guide your reasoning.
    Your ultimate goal is to select the most appropriate answer.:
Below is the template for the schema you need to fill out:
Broad Category:
Identify the overarching subject and general category to which the
    problem belongs.
Refinement:
Describe further details or specific aspects that narrow down the broad
    category.
Specific Scope:
Define the precise focus or context of the problem within the refined
    category.
Goal:
Clearly state the objective or intended outcome of solving the problem.
Finally, summarize the schema in a few sentences to help students grasp
    the key points. The problem you need to abstract is as follows:"""

User Messages:

[
  {
    "role": "user",
    "content": "Question: trans-cinnamaldehyde was treated with
    methylmagnesium bromide, forming product 1.\n\n1 was treated with
    pyridinium chlorochromate, forming product 2.\n\n3 was treated with (
    dimethyl(oxo)-l6-sulfaneylidene)methane in DMSO at elevated
    temperature, forming product 3.\n\nhow many carbon atoms are there in
     product 3?\nCandidates: \n1. 14\n2. 12\n3. 10\n4. 11"
  },
  {
    "role": "assistant",
    "content": "#### Schema:\n##### broad_category:\nOrganic Chemistry \
    u2192 Reaction Mechanisms \u2192 Organic Synthesis\n\n#####
    refinement:\nThis problem involves a series of organic reactions
    starting from trans-cinnamaldehyde and leading to a final product
    through specific reagents and conditions.\n\n##### specific_scope:\
    nThe focus is on determining the number of carbon atoms in the final
    product (product 3) after a sequence of reactions involving trans-
    cinnamaldehyde, methylmagnesium bromide, pyridinium chlorochromate,
    and a sulfaneylidene compound.\n\n##### goal:\nCalculate the total
    number of carbon atoms in product 3 after completing the series of
    reactions.\n\n#### Summary:\nThis problem involves a series of
    organic reactions that transform trans-cinnamaldehyde into a final
    product through various reagents and conditions. The task is to track
     the carbon atoms throughout the reactions to determine the total
    number in the final product (product 3). By analyzing the structure
    of each intermediate and the final product, we can ascertain the
    number of carbon atoms present. The goal is to arrive at the correct
    count of carbon atoms in product 3."
  },
  {
    "role": "user",
    "content": "Now that you've seen a similar question along with its
    schema and solution, reflect on \n    how you could refine or improve
     your previously generated schema for the **current** question. Focus
     on alignment \n    in categories, scope specificity, and consistency
     of abstraction.\n## Question:\nQuestion: Trans-cinnamaldehyde
    undergoes a reaction with methylmagnesium bromide to yield compound
    1. Compound 1 is then oxidized using pyridinium chlorochromate to
    form compound 2. Subsequently, compound 2 reacts with (dimethyl(oxo)
    -\u03bb6-sulfaneylidene)methane in DMSO at high temperature to
    produce compound 3. How many carbon atoms are present in compound 3?\
    nCandidates: \n1. 12\n2. 10\n3. 9\n4. 11\n          \n### Answer to
    this question is: 11\n\n### Below is the schema and summary of this
```

```
    question: \n#### Schema:\n##### broad_category:\nChemistry \u2192
    Organic Chemistry \u2192 Reaction Mechanisms\n\n##### refinement:\
    nThis problem involves a series of organic reactions including
    Grignard reactions, oxidation, and a reaction with a sulfonyl
    compound.\n\n##### specific_scope:\nThe focus is on determining the
    number of carbon atoms in the final product (compound 3) after a
    sequence of reactions starting from trans-cinnamaldehyde.\n\n#####
    goal:\nCalculate the total number of carbon atoms in compound 3 after
     all specified reactions have been completed.\n\n#### Summary:\nThis
    problem involves a series of organic reactions that transform trans-
    cinnamaldehyde through various steps, including a Grignard reaction,
    oxidation, and a reaction with a sulfonyl compound. The objective is
    to determine the number of carbon atoms in the final product,
    compound 3. By analyzing the reactions and counting the carbon atoms
    at each stage, we find that compound 3 contains 11 carbon atoms."
  },
  {
    "role": "assistant",
    "content": "#### Schema:\n##### broad_category:\nOrganic Chemistry \
    u2192 Synthetic Transformations \u2192 Carbon Skeleton Accounting\n\n
    ##### refinement:\nMulti-step sequence featuring: (1) Grignard
    addition to an \u03b1,\u03b2-unsaturated aldehyde (trans-
    cinnamaldehyde), (2) PCC oxidation of the resulting alcohol to a
    ketone, and (3) Corey\u2013Chaykovsky-type methylene transfer using
    dimethylsulfoxonium methylide in DMSO at elevated temperature.
    Emphasis on how each step affects carbon count.\n\n#####
    specific_scope:\nTrack carbon atoms starting from trans-
    cinnamaldehyde (C9). Step 1: MeMgBr adds a methyl to the carbonyl
    carbon (allylic secondary alcohol) \u2192 +1 carbon. Step 2: PCC
    oxidizes alcohol to ketone \u2192 0 carbon change. Step 3: Dimethyl(
    oxo)-\u03bb6-sulfaneylidene)methane (dimethylsulfoxonium methylide)
    transfers a methylene to the carbonyl (Corey\u2013Chaykovsky
    epoxidation/methylenation) \u2192 +1 carbon. Determine the total
    carbons in product 3.\n\n##### goal:\nUse stepwise carbon accounting
    across the reaction sequence to identify the number of carbon atoms
    in product 3 and select the correct option.\n\n#### Summary:\nThis is
     a carbon-tracking problem across a three-step synthesis from trans-
    cinnamaldehyde. Only the Grignard addition and the sulfur ylide
    methylene transfer change the carbon count, each adding one carbon;
    the PCC oxidation does not. Applying this schema leads to the correct
     carbon total in the final product."
  },
  {
    "role": "user",
    "content": "Now that you've refined your schema for the current
    question, select the most appropriate answer. "
  }
]
```

## F.2 ONE-SHOT

System Message:

```
"""Select the most appropriate answer. You will be shown a sample
    question with its correct answer.
Then, answer a new question of the same type."""
```

User Messages:

```
[
  {
    "role": "user",
    "content": "Example Question:\nQuestion: Trans-cinnamaldehyde
    undergoes a reaction with methylmagnesium bromide to yield compound
```

```
      1. Compound 1 is then oxidized using pyridinium chlorochromate to
      form compound 2. Subsequently, compound 2 reacts with (dimethyl(oxo)
      -\u03bb6-sulfaneylidene)methane in DMSO at high temperature to
      produce compound 3. How many carbon atoms are present in compound 3?\
      nCandidates: \n1. 12\n2. 10\n3. 9\n4. 11\nAnswer: 11"
    },
    {
      "role": "user",
      "content": "Now try a similar question:\nQuestion: trans-
      cinnamaldehyde was treated with methylmagnesium bromide, forming
      product 1.\n\n1 was treated with pyridinium chlorochromate, forming
      product 2.\n\n3 was treated with (dimethyl(oxo)-l6-sulfaneylidene)
      methane in DMSO at elevated temperature, forming product 3.\n\nhow
      many carbon atoms are there in product 3?\nCandidates: \n1. 14\n2.
      12\n3. 10\n4. 11"
    }
]
```

## F.3  ONE-SHOT + COT

System Message:

```
"""Select the most appropriate answer. You will be shown a sample
    question with its correct answer.
Then, answer a new question of the same type."""
```

User Messages:

```
[
  {
    "role": "user",
    "content": "Example Question:\nQuestion: Trans-cinnamaldehyde
    undergoes a reaction with methylmagnesium bromide to yield compound
    1. Compound 1 is then oxidized using pyridinium chlorochromate to
    form compound 2. Subsequently, compound 2 reacts with (dimethyl(oxo)
    -\u03bb6-sulfaneylidene)methane in DMSO at high temperature to
    produce compound 3. How many carbon atoms are present in compound 3?\
    nCandidates: \n1. 12\n2. 10\n3. 9\n4. 11\nAnswer: 11"
  },
  {
    "role": "user",
    "content": "Now try a similar question:\nQuestion: trans-
    cinnamaldehyde was treated with methylmagnesium bromide, forming
    product 1.\n\n1 was treated with pyridinium chlorochromate, forming
    product 2.\n\n3 was treated with (dimethyl(oxo)-l6-sulfaneylidene)
    methane in DMSO at elevated temperature, forming product 3.\n\nhow
    many carbon atoms are there in product 3?\nCandidates: \n1. 14\n2.
    12\n3. 10\n4. 11\nPlease think step by step."
  }
]
```

## G  GPT-5'S PERFORMANCE ON HUMANITY'S LAST EXAM

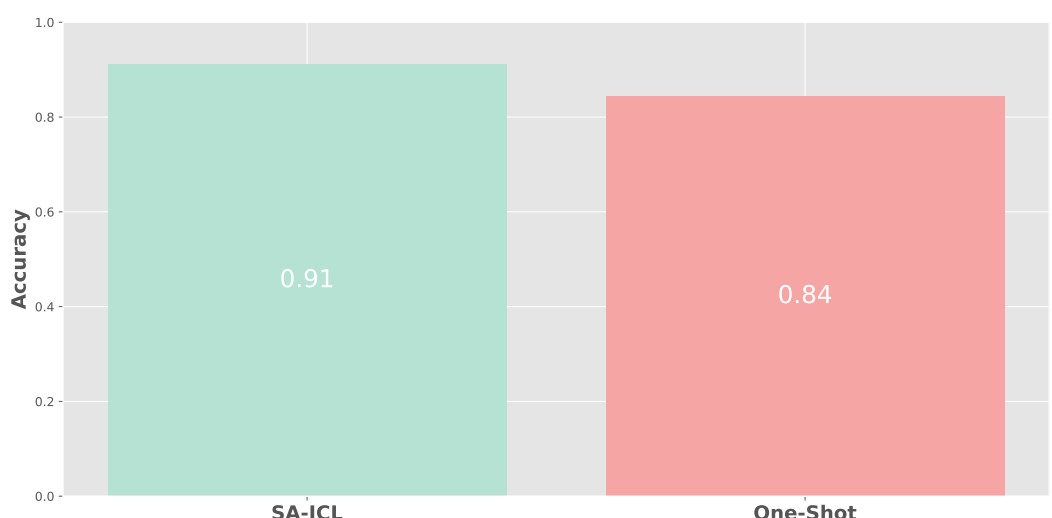

Figure G.2: Performance of GPT-5 on the chemistry, physics, biology multi-choice questions from the Humanity's Last Exam (HLE) benchmark dataset (Phan et al., 2025) (N=191, we filtered out questions with images), when provided with the *Essentially Same* examples. We generated the synthetic data as well using the same procedures described in Section C.

When GPT-5 was introduced, we used the model to run the same experiments. However, we realized that when provided with *Essentially Same* examples, GPT-5 reached near 100% accuracy in the GPQA dataset, eliminating the need for explicit reasoning modules introduced by SA-ICL. Therefore, we used a more challenging dataset, Humanity's Last Exam (HLE), to repeat the same experiments using *Essentially Same* examples, and saw that SA-ICL results in a 7% improvement in accuracy compared to standard One-Shot. This gain is particularly notable given the already strong baseline accuracy of 84%.

Note that this subset of HLE includes 136 biology and medicine questions, 34 physics questions, and 21 chemistry questions. We acknowledge that most of the questions are in the biology and medicine sub-categories, which is a domain that was not tested in the main experiments shown in Section 5.3.

# H INTERPRETABILITY EXPERIMENTS

## H.1 REPORT ON TOKEN COUNTS AND CORRECTNESS FOR THE FIRST 10 *GPQA-Chemistry* QUESTIONS

The interpretability experiments are conducted via logit-centric methods to get the overall confidence of LLM generation by retrieving the token-level log likelihood (Zhang et al., 2025) from the OpenAI chat completions API for each of our tested ICL prompting strategies. We implemented a customized chat interface to display the log-probability in colored texts using the color schema provided in the legend in Figure H.3 for better visualization and interaction.

From Table 3, we observed that SA-ICL achieves the highest accuracy in these questions while simultaneously generating fewer reasoning tokens compared to One-Shot + CoT, except for the sixth question.

## H.2 VISUALIZATION OF THE LOG-PROBABILITIES OF EACH TOKEN IN THE SOLVER'S OUTPUT

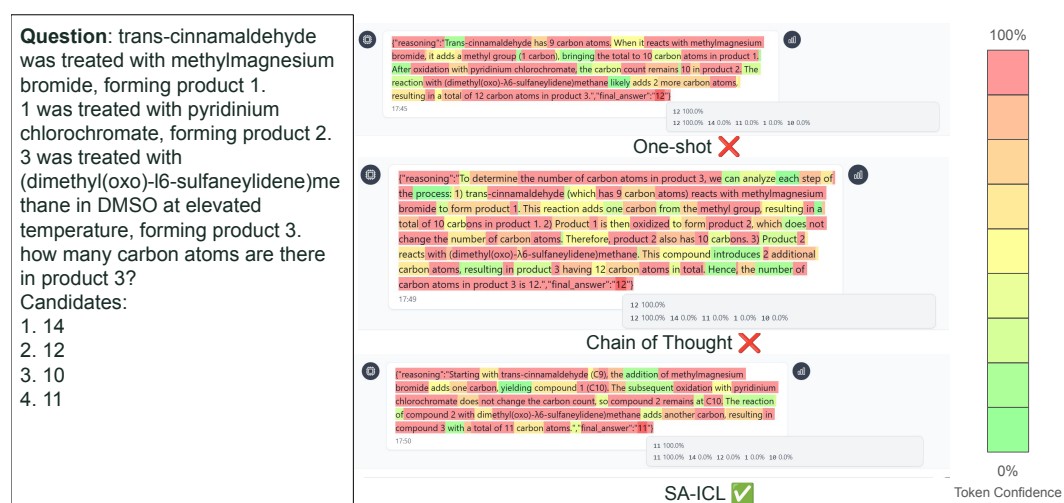

Figure H.3: The probabilities of each token in the problem solver output. Note that only SA-ICL confidently gave the correct answer for this question, while pure One-Shot or One-Shot + CoT confidently gave the incorrect answer.

# I   ABLATION STUDY ON THE SCHEMA ACTIVATION

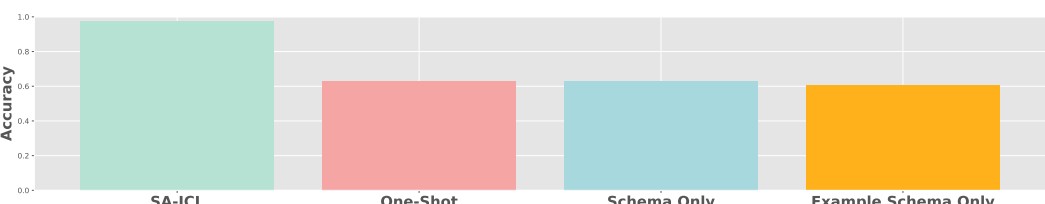

Figure I.4: GPT-4o Mini solving the *GPQA-Physics* questions with examples that are *Essentially Same*.

To further investigate in isolation the effectiveness of schema activation within our framework, we conducted a controlled ablation study comparing four distinct conditions: SA-ICL, One-Shot, Schema Only, and Example Schema only.

## I.1   DESIGN AND NATURE OF ABLATION STUDY

This ablation analysis was designed to determine whether the performance improvements observed in SA-ICL stem from the schema activation mechanism itself or merely from the presence of abstracted examples. In particular, we are interested in the case when prior examples are of high quality, and therefore conducted this study by using GPT-4o Mini to solve questions in the *GPQA-Physics* subset with prior examples that are *Essentially Same* in terms of synthetic similarity. The Schema Only condition provided models with abstract problem schemas without corresponding episodic examples, while Example Schema Only presented abstracted schemas derived from examples but without the schema activation process that is integral to the full SA-ICL procedure.

## I.2   INTERPRETATION OF ABLATION RESULTS

Figure I.4 presents the comparative accuracy results across all four scenarios. The complete SA-ICL framework demonstrated clear superiority over the other three approaches, with a near 40% accuracy boost over the other three conditions. In fact, neither Schema Only nor Example Schema Only result in a clear benefit over One-Shot, which acts as a baseline for this ablation study. This observation provides direct evidence that the schema activation mechanism constitutes the critical component driving the effectiveness of our in-context learning framework.

## I.3   SCHEMA DORMANCY

The results of our ablation study strongly suggest that the mere presence of an abstracted schema within the prompt context is insufficient to improve model performance. The sheer significance of schema activation leads to a phenomenon which we coin *schema dormancy*. This term describes a state where a refined schema tailored to a specific task only exists as passive surface-level contextual information rather than an active cognitive framework that guides the language model's reasoning process. When the explicit activation step of the SA-ICL framework is omitted — as observed in the Schema Only and Example Schema Only conditions — the LLM fails to effectively integrate the abstracted reasoning structure into its problem-solving approach. This finding highlights that language models do not implicitly adopt and utilize abstract schemas without an explicit technique to activate this otherwise dormant state.

## THE USE OF LARGE LANGUAGE MODELS

The authors used large language models such as ChatGPT to assist with paper writing, mostly for polishing the text, and after which the authors did a thorough check to ensure the polished paper faithfully delivered the authors' messages.

