# OpenReview forum: "Schema for In-Context Learning"
_ICLR.cc/2026/Conference — ICLR 2026 Conference Withdrawn Submission_

### Official Review · Reviewer_1Fkr · 2025-10-28

**Soundness:** 2
**Presentation:** 3
**Contribution:** 2
**Rating:** 4
**Confidence:** 4

**Summary:**

The paper proposes SA-ICL, which enables large language models to form and activate abstract schemas from prior examples to guide reasoning on new tasks. This schema-based approach improves accuracy by up to 39.7% on scientific reasoning benchmarks while enhancing interpretability and efficiency compared to traditional example-driven ICL.

**Strengths:**

- The paper introduces a cognitively grounded framework, SA-ICL that models human schema activation, bridging cognitive psychology and machine learning.
- SA-ICL demonstrates substantial accuracy improvements (up to 39.67% in chemistry and 34.45% in physics) across multiple large language models.

**Weaknesses:**

- The experiments are restricted to scientific reasoning (physics and chemistry) on the GPQA dataset, leaving unclear whether SA-ICL generalizes to other reasoning domains such as commonsense, mathematics, or open-ended tasks.
- The model scope is also limited. It would be beneficial if the authors could expand to more models, such as the thinking models.
- Typos: such as "gicen" in line 932.
- The study primarily contrasts SA-ICL with One-Shot and CoT prompting, omitting comparisons to stronger retrieval-based or structured reasoning baselines.

**Questions:**

- I'm still concerned about the intellectual contributions of this paper, and how the proposed methods could been adopted in real applications.

---

### Official Review · Reviewer_jLfB · 2025-10-30

**Soundness:** 1
**Presentation:** 1
**Contribution:** 1
**Rating:** 2
**Confidence:** 4

**Summary:**

The paper introduces Schema Activated In Context Learning, a framework that augments example based prompting with explicit schemas that capture reusable reasoning patterns. Given a new problem, the method forms a representation, retrieves relevant schemas and past examples from memory, activates and adapts a chosen schema, and then uses it to guide inference. The memory links schemas with episodic traces so the system can associate a query with similar prior situations.

**Strengths:**

The human-inspired ICL is an interesting narrative. The discussion of schemas from cognitive science is inspiring.

**Weaknesses:**

There is no comparison with many-shot ICL. It remains unclear the influence of exemplars. The paper only uses one-shot ICL which is very uncommon for ICL.

I feel This work is not really about ICL. It is more like adding advanced system prompts to tell models what methods or tools to use to tackle the problem. The proposed framework is similar to an advanced prompting technique like self-refine [1] or reflexion [3]. The llms refine their strategy during prompting. There are also many advanced ICL frameworks (e.g., [2]). With that said, more baselines should be used as comparison. Currently the empirical support is weak.  And experiments are only evaluated on one scientific QA dataset. Evaluation on more general-domain data is needed.

Schema is not very clearly explained. Appendix E and F are not explicitly mentioned in a clear way to let readers understand schema is actually a prompting template. The current writing is unnecessarily complex.

The citation format seems wrong through the whole paper. there is no bracket.

The exemplar selection part is weak. Ablation study or multiple experiments on what exemplars are chosen can be helpful to test the robustness of Schema-ICL framework.

[1] Madaan, Aman, et al. "Self-refine: Iterative refinement with self-feedback." Advances in Neural Information Processing Systems 36 (2023): 46534-46594.

[2] Zhang, Yiming, Shi Feng, and Chenhao Tan. "Active example selection for in-context learning." arXiv preprint arXiv:2211.04486 (2022).

[3] Shinn, Noah, et al. "Reflexion: Language agents with verbal reinforcement learning." Advances in Neural Information Processing Systems 36 (2023): 8634-8652.

**Questions:**

What is S=R(x)? Is S hidden states or verbal prompts? How do you get R(x)?

---

### Official Review · Reviewer_JsMQ · 2025-11-01

**Soundness:** 1
**Presentation:** 2
**Contribution:** 1
**Rating:** 2
**Confidence:** 4

**Summary:**

The paper introduces Schema-Activated In-Context Learning (SA-ICL), a framework inspired by cognitive science schema theory. The approach extracts abstract "schemas" (structured templates of reasoning steps) from demonstration examples, retrieves relevant schemas for new problems, and uses schema activation to guide LLM reasoning. The authors evaluate SA-ICL on graduate-level chemistry and physics questions from the GPQA dataset, reporting improvements up to 39.67% over one-shot prompting when high-quality similar examples are available. The method aims to bridge example-driven and abstraction-driven in-context learning approaches.

**Strengths:**

1. The paper contributes to the field of prompting engineering by proposing a new schema-based ICL framework with five stages: problem representation, schema retrieval, episodic retrieval, schema activation, and schema-guided solving. The method could be practically relevant to the recent emerging topic of vibe coding with LLMs.
2. The paper makes a genuine attempt to bridge cognitive science (schema theory) and machine learning, which is relatively underexplored in the ICL literature. While structured prompting and retrieval-augmented methods exist, the explicit operationalization of schema activation that distinguishes between retrieval and activation, and incorporates assimilation/accommodation concepts, could offer an interesting perspective.
3. While the scope is limited and the baselines are weak, the overall experimental design covers multiple levels of analysis and spans across multiple model families.
4. The paper has a generally clear presentation with good organization.

**Weaknesses:**

1. The evaluation is confined to only two domains (chemistry and physics) on a single benchmark (GPQA), testing exclusively on closed-ended multiple-choice questions. This limited scope is insufficient to support the broad claims made about advancing in-context learning. There are no evaluations on standard ICL benchmarks (e.g., tasks from BBH, BB-Extra-Hard, MMLU-pro subsets), and also hard reasoning benchmarks like AIME, MedXpertQA, and ZebraLogic.
2. There are no open-ended generation tasks where schema benefits could be more meaningful. The single GPT-5 experiment on Humanity's Last Exam (Appendix G) is too limited and confounded by domain distribution (136/191 questions are biology/medicine, which were not tested in the main experiments)
3. The paper compares primarily against vanilla One-Shot and basic CoT, missing important classic and recent structured reasoning methods. First of all, the most obvious baseline, Few-shot learning (3-shot, 5-shot), which is standard in ICL literature, is missing. Other missing baselines include: Self-Consistency with multiple reasoning paths, Unified Self-Consistency, Reflexion, ReAct, Tree-of-thoughts, Program-of-Thoughts, MetaICL, and Self-Refine.
4. There is no analysis of computational costs despite the method requiring multiple LLM calls per query. The paper should report wall-clock time, token counts, and estimated API costs. Compare cost-performance trade-offs (e.g., SA-ICL with 1 example vs. vanilla 5-shot). Analyze whether performance improvements justify additional costs. The authors should also discuss practical deployment considerations.
5. The schema template (Appendix E) appears hand-crafted with four specific components (Broad Category, Refinement, Specific Scope, Goal), but no ablation studies examine alternative designs. See more details in the Questions section.

**Questions:**

1. How sensitive are results to the specific template structure?
2. Would different schema components work better?
3. Is this template optimal for chemistry/physics or domain-agnostic?
4. Can the template be learned or adapted?

---

### Official Review · Reviewer_8TP9 · 2025-11-03

**Soundness:** 2
**Presentation:** 3
**Contribution:** 2
**Rating:** 4
**Confidence:** 4

**Summary:**

This paper introduces Schema-Activated In-Context Learning (SA-ICL), a novel and framework inspired by the "schema" concept from cognitive science. The authors argue that in existing few-shot in-context learning, models only learn surface-level "pattern matching" and fail to acquire high-level, structured abstract features. To address this, the proposed SA-ICL method explicitly guides the model to extract the problem's structural information (the "schema"), indexes similar schemas and retrieves specific episodic examples from a memory bank, and finally uses these retrieved elements to refine (or "activate") the original schema. Experiments show that SA-ICL significantly outperforms the standard one-shot method on the GPQA dataset. Furthermore, extensive ablation studies reveal that LLMs lack the intrinsic ability to perform schema activation and require explicit guidance to do so.

**Strengths:**

1. Addressing the view that one-shot ICL is merely pattern matching, this paper innovatively introduces the "schema" concept from cognitive science. By explicitly guiding the LLM to extract high-level abstractions of a problem, it achieves significantly better results than standard one-shot.
  2. The internal experiments (ablation studies) are thorough. What I find most impressive is the discovery of the "schema dormancy" phenomenon: LLMs do not spontaneously utilize schemas but require explicit activation. This is a point rarely discussed in prior work.
  3. This paper seeks to explore an alternative path for ICL, introducing novel perspectives to this problem, which is commendable.

**Weaknesses:**

1. The biggest issue is the use of overly weak baselines, an unconvincing experimental design, and a singular dataset. The main experiments are largely devoted to arguing the effectiveness of "schema" versus the "one-shot" method. However, a crucial end-to-end comparison against mainstream methods on the GPQA dataset, which is the most important part, appears to be absent,  This omission casts doubt on the practical utility of the SA-ICL framework.
   2. Several aspects of the paper are not clearly articulated, mainly regarding the methodology and experiments. Please see my detailed questions below.

**Questions:**

While this paper novelly introduces the "schema" concept in an attempt to align ICL with human cognitive processes and pave a new research path, I have some concerns regarding this type of pioneering work.

1. Given the extensive literature reviewed in the Related Work, why are the primary baselines limited to only One-Shot and One-Shot+CoT? A paper that claims to "pave a new path" should provide a comprehensive experimental comparison against current mainstream methods, including an analysis of the performance-efficiency trade-off. Without such support, it is difficult to convince researchers to follow this work.

2. Even in the comparison against One-Shot, the paper lacks an end-to-end (E2E) evaluation across multiple datasets. It appears that to isolate the effectiveness of the "schema" concept, the authors have fed a pre-selected, high (low)-quality example to the model for each query, rather than letting the system retrieve it. This functions more as a fine-grained ablation study, while the practical, E2E utility of the SA-ICL framework (Algorithm 1) is left unproven.

3. The paper claims SA-ICL is an efficient and lightweight method. While Table 1 shows it is *effective* (high example-utilizing rate), no experiment supports its *efficiency* (lightweight). In practice, the full pipeline (e.g., Appendix F.1) seems to require at least three LLM calls (initial schema generation, schema activation, and final problem-solving). This suggests its computational cost is significantly higher than One-Shot. Could the authors clarify whether the token counts reported in Table 2 represent only the *final* reasoning step, or the *total* tokens consumed across the entire SA-ICL process?

4. The paper criticizes CoT as being instance-specific. If the authors claim that schemas offer cross-domain generalizability, this claim should be substantiated by testing on multiple, diverse datasets beyond GPQA. At a minimum, a mathematics dataset should be added. Furthermore, should the method also be tested on non-logic-intensive datasets (e.g., commonsense reasoning) to demonstrate the breadth of its applicability?
5. The formalization in Appendix A.1 introduces an association weight $w_{ij}(t)$ with a decay function. Does this imply that, in a practical deployment, the memory bank is dynamic and expected to be updated over time? This aspect of the framework is not explored in the experiments.

6. In the main comparison (e.g., Figure 2, Table 1), SA-ICL is compared against a standard One-Shot baseline, which is given only an example (Q+A). However, the SA-ICL method is given both the example and its schema, which acts as an abstracted reasoning trace. For a fairer comparison, shouldn't the baseline also be provided with a reasoning trace?

7. SA-ICL is presented as a retrieval mechanism operating at an abstract level, built on top of the RAG paradigm. Is there experimental support, in an E2E evaluation, showing that schema-based retrieval is better than query-based retrieval?

---

### Author Response · Authors · 2025-12-03

Thank you all the reviewers for all your great and valuable feedback, which we will take into account for our next revision of this paper. We have provided more results in a later comment.

---

### Author Response · Authors · 2025-12-03

We deeply **appreciate** the constructive feedback all the reviewers have been provided.

We **acknowledge** the limitations of the experiments we conducted, and would like to provide clarifications and share more experiment results to express our message to the community that abstraction-based schema learning could help LLMs, especially when the knowledge base is dense.

### Regarding the limitations of Datasets:
We ran our experiments with three more datasets: MedXpert, MMLUCollegeMath, and CommonsenseQA.

- When the knowledge is dense (“High”), SA-ICL still outperformed one-shot by at most 17.57%, and 5-shot by at most 9.46% from the MMLUCollegeMath benchmark. **This result is aligned with the experiments we presented in the paper, showing that SA-ICL is a stronger reasoning framework in a more generalized way.**

- While the results are promising, we **acknowledge** that we didn’t run the extended experiments with more models. We will attempt more models in the next revision of this paper.

| Dataset            | Method                     | Accuracy |
|--------------------|-----------------------------|----------|
| **MMLUCollegeMath** | SA-ICL                      | **0.7973** |
|                    | One-Shot                   | 0.6216   |
|                    | One-Shot with Reasoning    | 0.7162   |
|                    | 5-Shots                    | 0.7027   |
||||
| **MedXpertQA**      | SA-ICL                      | **0.7342** |
|                    | One-Shot                   | 0.6034   |
|                    | One-Shot with Reasoning    | 0.6667   |
|                    | 5-Shots                    | 0.7027   |
||||
| **CommonSense**     | SA-ICL                      | **0.9200** |
|                    | One-Shot                   | 0.9000   |
|                    | One-Shot with Reasoning    | 0.8400   |
|                    | 5-Shots                    | 0.8800   |



### Regarding the limitations of Baselines:

- We enhanced one-shot with reasoning paths (generated by GPT-4o with ground truths). We re-ran the experiments using GPT4o-mini with one-shot in the extended datasets, and we noticed that one-shot with reasoning path still underperformed SA-ICL in MedXPert, U-Math-Pro (we were actually using the original MMLU questions), and CommonSenseQA.

- We also implemented Multi-Shot and ran 5-Shot as mentioned above. We acknowledge that the baseline pool is still pretty small. Working with more baselines will make our model more convincing for real-world usage. However, since abstraction-based schema learning is a novel attempt to incorporate human’s reasoning behaviour grounded on a foundational cognitive science theory, we wanted to make our comparisons & evaluations straightforward: when given the best-ever example, could LLMs learn from the best example implicitly, or will LLMs benefit by mincing humans’ reasoning and learning? We believe that our experiments have shown useful insights.

- Although when the example is not high-quality enough, SA-ICL wasn’t as effective as it would have been with a dense knowledge base, we wanted to point out that *our schema template was not fine-tuned towards specific domains yet* and the framework relied on one example for one schema, was made the schema less generalizable. We believe that our work can inspire more future work to make abstraction-level reasoning as effective as it would be, even when the knowledge is not dense.

### Regarding there’s no end-to-end evaluation

- We apologize for the confusion. But we did end-to-end evaluation. Due to the nature of this study, we want to make sure that both conditions to be compared have exactly the same example(s) in the context. Therefore, we cached the retrieval processes, which might result in the misunderstanding that the example was hand-picked. All examples were selected by cos-similarity, labels (by GPT4o), or Cohere Reranker.

### Regarding the concerns with token usage

- We acknowledge that the token usage reported was the last layer only. We would make the wording clearer. We wanted to convey that our mechanism could reduce the number of output tokens from the last layer/api call, avoiding model overthinking and resulting in a higher performance.

- We also wanted acknowledge that our framework might be slower compared to one-shot, but we want to focus on the performance boost and send the LLM community a message that ***the abstractions (schemas) behind the examples (demonstrations) were what lead to a higher learning efficiency in the context without fine-tuning.*** We believe that the current paradigm of example-driven In-Context Learning might not be exploiting the example well.

---

### Note · Authors · 2026-01-15

**Comment:**

We thank the reviewers for their insightful and valuable feedback.

**Withdrawal Confirmation:**

I have read and agree with the venue's withdrawal policy on behalf of myself and my co-authors.